# Genomic Sequencing and Characterization of Two *Auricularia* Species from the Qinling Region: Insights into Evolutionary Dynamics and Secondary Metabolite Potential

**DOI:** 10.3390/jof11050395

**Published:** 2025-05-20

**Authors:** Jianzhao Qi, Shijie Kang, Ming Zhang, Shen Qi, Yulai Li, Khassanov Vadim, Shuangtian Du, Minglei Li

**Affiliations:** 1Shaanxi Key Laboratory of Natural Products & Chemical Biology, College of Chemistry & Pharmacy, Northwest A&F University, Yangling, Xianyang 712100, China; 2Center of Edible Fungi, Northwest A&F University, Yangling, Xianyang 712100, China; 3School of Soil and Water Conservation Science and Engineering, Northwest A&F University, Yangling, Xianyang 712100, China; 4Department of Plant Protection and Quarantine, Faculty of Agronomy, S. Seifullin Kazakh Agrotechnical University, Zhenis Avenue, Astana 010011, Kazakhstan

**Keywords:** *Auricularia* spp., genome sequencing, Qinling region

## Abstract

*Auricularia* mushrooms, common bulk edible fungi, have considerable culinary and medicinal value. The Qinling region, represented by Zhashui County, is the main production area of *Auricularia* mushrooms in China. In this study, two wild *Auricularia* strains, M12 and M13, selected from the Qinling region for their desirable horticultural traits after domestication, were sequenced and characterized. Sequencing assembly results based on Illumina NovaSeq and PacBio Sequel II HiFi showed that the M12 genome was 56.04 Mbp in size, with 2.58% heterozygosity and 14.13% repetitive sequences, and was anchored on 12 chromosomes using HI-C technology. In contrast, the M13 genome was 52.10 Mbp, showed 2.34% heterozygosity, 13.89% repetitive sequences, and was assembled into 12 scaffolds. Collinearity analysis revealed extensive homologous regions between the M12 and M13 genomes. Phylogenetic analysis suggested that the divergence between M12 and M13 occurred approximately 4.575 million years ago (MYAs), while their divergence from *Auricularia subglabra* TFB-10046 SS5 occurred approximately 33.537 MYAs. Analyses of CYP450, carbohydrate-active enzymes (CAZymes), and gene family expansion/contraction revealed distinct genomic features between the two strains. SSR and LTR insertion time analyses revealed the genome dynamics of the two strains during their evolution. Analysis of secondary metabolite-associated biosynthetic gene clusters (BGCs) provides powerful clues to understand the origin of bioactive compounds in the *Auricularia* mushroom. This work represents the first genome sequencing of the *Auricularia* species derived from the Qinling region. These results not only enriched our understanding of the *Auricularia* genome but also provided an important genomic resource and theoretical basis for the subsequent genetic breeding, functional gene mining, and development of medicinal components of *Auricularia* species.

## 1. Introduction

The genus *Auricularia*, part of the family Auriculariales within the class Basidiomycota, is a major group that currently contains nearly 40 identified species [1]. These species are commonly parasitic on the stems of angiosperms and occasionally on gymnosperms, with a wide global distribution [2,3]. The fruiting bodies of *Auricularia* species are gelatinous and ear-shaped, hence the name “wood ear” mushrooms. The genus *Auricularia* was first described in 1780 [4], and two species were introduced in 1822 [5], namely *Auricularia mesenterica* (which later became the type species) and *Auricularia sambuci* (a synonym for *Auricularia auricula-judae*). The name *Auricularia auricula-judae* was first proposed in 1789 [6]. In China and other East Asian countries, several members of the genus *Auricularia* have a long history of consumption and medicinal use, with [3,6]. Recent advances in molecular biology have shown that the widely distributed and cultivated variety in China, known as Heimuer (黑木耳), is a new species within the genus, named *Auricularia heimuer* F. Wu, B.K. Cui, and Y.C. Dai [6]. Another important group among the wild and cultivated *Auricularia* species in China is Maomuer (毛木耳) [3], which, according to molecular identification, consists mainly of two distinct *Auricularia* species: *A. cornea* and *A. polytricha*. The latter was first defined as a new species in Jamaica and later confirmed as a synonym of *A. nigricans* (Sw.) Birkebak [7]. *Auricularia cornea* is an interesting species within the genus, with brown or white fruiting bodies, cultivated in regions such as China [8,9], Thailand [10], and Nigeria [11]. A recently cultivated pink variety of *A. cornea* in Sichuan Province, China, is not only ornamental but also nutritious [8,9]. Statistical data for 2023 show that China’s production of *A. heimuer* reached 7.1447 million tons, and that of Maomuer (*A. cornea* and *A. polytricha*) was 2.8522 million tons, ranking second and fourth, respectively, in China’s edible fungi production. These data highlight the significant position of *Auricularia* species in China’s edible fungus industry.

The *Auricularia* mushroom, a highly prized medicinal and culinary fungus in China, is renowned not only for its culinary value but also for its significant medicinal benefits. Dating back to the Eastern Zhou period, some 2000 years ago, Heimuer (*A. heimuer*) was already a delicacy for the royal court [12]. Traditional Chinese medicine holds the medicinal value of Heimuer (*A. heimuer*) in high regard, with the earliest records found in the Ming Dynasty’s masterpiece, “Compendium of Materia Medica” [13]. Modern pharmacological research has further confirmed the bioactivity and pharmacological effects of polysaccharides from *Auricularia* fungi. Specifically, crude polysaccharides extracted from *A. auricula-judae* exhibit antimicrobial and antioxidant properties [14], while its water-soluble polysaccharides have been proven to lower blood sugar levels [15]. Moreover, the *A. auricula-judae* polysaccharides can alleviate symptoms of obesity in mice [16], and their acid hydrolysates also improve glucose metabolism in diabetic mice [17]. Studies indicate that *A. auricula* polysaccharides mitigate obesity by modulating the gut microbiota [18]. Additionally, the water-soluble polysaccharides of *A. polytricha* have been found to have the potential to reduce cholesterol levels [19]. These findings further substantiate the value of *Auricularia* fungi in the realms of health and nutrition.

The Qinling region is one of the principal production areas for *Auricularia* mushrooms in China. The cultivation of *Auricularia* in this region has a long history and is renowned for its high-quality produce and unique natural cultivation environment. The southern foothills of the Qinling Mountains, particularly in the southern part of Shaanxi Province, including Zhashui County, are characterized by a humid climate and ample sunlight, which are highly conducive to the growth and reproduction of *Auricularia* mushrooms. The Qinling region is home to a rich diversity of wild edible fungi, including *Auricularia* species. Local inhabitants have consumed and cultivated *Auricularia* mushrooms for over two thousand years. With the implementation of China’s poverty alleviation policies, the *Auricularia* industry in the Qinling region has seen significant development. Over the past decade, during our in-depth surveys of wild *Auricularia* germplasm resources in the Qinling region, we have collected several wild *Auricularia* varieties. Since June 2012, when *A. subglabra* became the first species in the genus *Auricularia* to have its genome sequenced [20], the genomes of nearly ten *Auricularia* species, including *A. heimuer* [21] and *Auricularia cornea* [22], have been successively published. However, no genetic information, including genomic data, has been reported for wild *Auricularia* species from the Qinling region. This lack of information has hindered the development of *Auricularia* resources in the Qinling Mountains. Therefore, in this study, we conducted genome sequencing of two wild *Auricularia* strains from the Qinling region that exhibit excellent horticultural traits. Based on the chromosome-level assembly, we investigated their evolutionary status and changes in gene families, performed comparative genomic analyses within the genus *Auricularia*, and conducted bioinformatics analyses of characteristic functional genes, including CAZymes, P450, and core genes involved in the biosynthesis of secondary metabolites.

## 2. Materials and Methods

### 2.1. Fungal Material and Nucleic Acid Extraction

The strains of *Auricularia* M12 and M13 are preserved at the Edible Fungi Centre of Northwest A&F University using the paraffin sealing method, with the respective designations M12 and M13. They are activated once every six months to ensure the stability of the strains. The mycelium used for DNA extraction was prepared by cultivating the original strains of the *Auricularia* M12 and M13 in PDB medium at 26 °C with 180 rpm agitation for five days. Genomic DNA was extracted using the NuClean Plant Genomic DNA Kit (CWBIO, Beijing, China) and checked for RNA contamination via agarose gel electrophoresis, and the purity of DNA (OD260/280 = 1.8–2.0) was assessed using a Nanodrop spectrophotometer.

For library construction, DNA samples with an amount ≥ 5 μg and a concentration ≥ 20 ng/μL were used. Purified genomic DNA was randomly fragmented into 300 bp segments using a Covaris sonicator. Fragmented DNA was repaired and end-prepared, and adapters were ligated using the NEBNext^®^ Ultra™ DNA Library Prep Kit for Illumina (NEB, San Diego, CA, USA) to create a DNA library. After quality control, the DNA library was sequenced on the Illumina NovaSeq high-throughput sequencing platform.

Hi-C Sequencing: Chromatin was first cross-linked with formaldehyde, and the genomic DNA was then digested with HindIII. A Hi-C library with an insert size of 300–700 bp was constructed. The library concentration and insert size were measured using Qubit 2.0 (Life Technologies (Thermo Fisher), Carlsbad, CA, USA) and Agilent 2100 (Agilent Technologies, Santa Clara, CA, USA), respectively, and the effective concentration was quantified using qPCR. After quality control, high-throughput sequencing was performed on the Illumina NovaSeq 6000 platform (San Diego, CA, USA) with a read length of PE150.

Nanopore Sequencing: DNA samples with a concentration ≥ 80 ng/μL and an amount ≥ 10 μg were used for library construction. Qualified genomic DNA was randomly fragmented using a Megaruptor (Diagenode, Denville, NJ, USA), and large DNA fragments were purified and enriched using magnetic beads. Large DNA fragment libraries were separated and recovered using the BluePippin automated nucleic acid purification system. The SQK-LSK109 sequencing kit (Oxford Nanopore Technologies, Oxford, UK) was used to prepare the sequencing library, following the manufacturer’s instructions. Fragmented genomic DNA was repaired and end-prepared, adapters were ligated, and the library was purified. The DNA library was quantified using Qubit. The constructed library was loaded onto Flow cells and sequenced on the Oxford Nanopore PromethION sequencer (Oxford Nanopore Technologies, Oxford, UK) for real-time single-molecule sequencing.

### 2.2. Genome Assembly and Chromosome Assembly

The genome size and complexity were estimated based on the k-mer distribution of Illumina short reads. The 21-mer distribution was statistically analyzed using GenomeScope 2.0 [23] with default parameters.

The initial assembly was performed using the NECAT (version 20200119) [24] software for genome error correction and scaffolding. The assembly was then corrected using Racon v1.4.1 software based on third-generation sequencing data for two rounds. Finally, two rounds of Pilon v1.23 [25] correction using second-generation reads were performed to remove heterozygosity and obtain the final assembly. The completeness of the genome and gene annotations was assessed using CEGMA 2.5 [26] and BUSCO 4.0 (Fungi Odb10 dataset) [27] with default parameters. To generate high-quality Hi-C reads for chromosome-level assembly, adapter sequences and low-quality paired-end (PE) reads were first removed [28]. Subsequently, high-quality Hi-C data were truncated at the presumed Hi-C ligation sites and aligned to the contigs using BWA (version 0.7.10-r789) [29]. Effective reads were filtered using HiC-Pro (version 2.10.0) [30], and only uniquely aligned PE reads were selected for further analysis. The genome sequences were clustered and ordered onto chromosomes using LACHESIS [31].

### 2.3. Genome Annotation

For the annotation of protein-coding genes, a combination of de novo prediction, homology search, and transcript-based annotation methods was used. De novo gene models were predicted using Augustus (version 3.1.0) [32]. In the homology-based approach, GeMoMa (version 1.7) was used to construct reference gene models, utilizing data from multiple *Auricularia* species, including *Auricularia subglabra* TFB-10046 SS5 and *Auricularia heimuer*. For transcript-based prediction, RNA-seq data were mapped to the reference genome using Hisat (version 2.1.0) and assembled using Stringtie (version 2.1.4). GeneMarkS-T (version 5.1) was used for gene prediction based on assembled transcripts. PASA (version 2.4.1) was employed to predict genes from a library composed of full-length PacBio transcripts and unigenes assembled by Trinity (version 2.11). Finally, gene models obtained from different methods were integrated using EVM (version 1.1.1) to complete the annotation of coding genes.

Functional annotation of protein-coding genes was performed by BLASTP alignment against multiple public databases, including EggNOG (http://eggnog5.embl.de/#/app/home.de/download/eggnog_5.0/ (accessed on 12 September 2024)), GO (http://geneontology.org), KOG, Pfam, TrEMBL, NCBI NR (http://ftp.ncbi.nlm.nih.gov/blast/db (accessed on 12 September 2024)), SwissProt (http://ftp.ebi.ac.uk/pub/databases/swissprot (accessed on 12 September 2024)), and KEGG (20191220 version), with an E-value cutoff of 1.0 × 10^−5^.

Non-coding genes such as miRNA, rRNA, tRNA, snoRNA, and snRNA were predicted using the following software: tRNA was predicted using tRNAscan-SE (version 1.3.1) with the parameter “eukaryote”. rRNA was predicted using Barrnap (version 0.9). miRNA was predicted using miRBase 21. snoRNA and snRNA genes were predicted based on the Rfam (version 14.5) database using INFERNAL (version 1.1) [33]. Pseudogenes were predicted using GenBlastA (version 1.0.4) and GeneWise (version 2.4.1).

The genes encoding CAZymes of 29 representative basidiomycetes, including *Auricularia* sp. Qinling M12 and *Auricularia* sp. Qinling M13 were annotated and classified using the CAZy database [34] (http://bcb.unl.edu/dbCAN2 (accessed on 12 September 2024)) via the HMMER package [35] (v 3.2.1, filter parameter E-value < 1 × 10^−5^; coverage > 0.35).

### 2.4. Comparative Genomic Analysis

The collinearity analysis of the genomes of M12 and M13 was performed using Diamond 2.0.4, and MCScanX (https://github.com/wyp1125/MCScanX (accessed on 4 March 2025)) [36] was used to identify syntenic blocks. The syntenic blocks were then visualized using SynVisio (https://synvisio.github.io/#/ (accessed on 5 March 2025)) [37] with the Chart Configuration set to Single-Level Analysis and Parallel Link Plot. *A. subglabra* TFB-10046 SS5, whose genome annotation information was available, was selected for comparative genomic analysis with M12 and M13. ClusterProfiler 3.6.0 [38] was used for GO and KEGG enrichment analysis of the positively selected genes of the two *Auricularia* species.

### 2.5. Single-Nucleotide Polymorphism (SNP) Site Detection

In this study, the second-generation sequencing data (FASTQ format) and the corresponding assembled genome files of two strains, M12 and M13, were used as input files. The reference genomes were first indexed using the BWA v0.7.17 software with the IS algorithm to establish an indexing system, generating index files with the suffixes .amb, .ann, .bwt, .pac, and .fai. Based on the generated index files, the bipartite sequencing reads were compared to the reference genome using the BWA-MEM algorithm to generate comparison files in SAM format [39].

Subsequently, the raw comparison files were formatted and quality controlled using SAMtools v1.13, which compressed the SAM files into binary BAM format while applying a filter criterion of Q-value ≥ 30 to reject low-quality reads. To ensure compatibility of subsequent analytical processes, genomic index files were created using the SAMtools faidx command. The GATK v4.3.0.0 [40] toolkit was also used for data preprocessing: the BAM files were coordinate sorted using the SortSam module, and the MarkDuplicates module was used to mark duplicate sequences and verify that the index files could be correctly recognized.

The SNP detection stage was performed using the GATK HaplotypeCaller module for variant identification to generate genomic variant format (GVCF) files. Variant sites were integrated by the GenotypeGVCFs module. Finally, the filtered SNP data were converted to MAP format using PLINK v1.9 [41], and the results were presented through the self-developed Python 3.10.0 visualization script.

### 2.6. Reconstruction of the Phylogenetic Tree

OrthoFinder v2.5.1 [42] was used to identify homologous genes in three *Auricularia* species, 25 representative basidiomycetes, and one ascomycete used as an outgroup. The protein sequences of single-copy orthologs were aligned using MAFFT v7.205 [43] with default parameters. Unique gene families were identified and annotated using the Pfam database. GO and KEGG enrichment analyses were performed using clusterProfiler v3.6.0. To estimate the best substitution model, ModelFinder was implemented in IQ-TREE 2.2.3 [44]. Maximum likelihood trees were constructed using RAxML 8.2 and the best-fitting substitution model, with 1000 bootstrap replicates. Divergence times were estimated using the MCMCTree program in PAML (version 4.9i) [45] with two calibration points obtained from the TimeTree database [46]: *Dacryopinax primogenitus* VS. *Calocera cornea* (76.8–94.1 MYA) and *Acaromyces ingoldii* VS. *Ustilago maydis* (211.4–370.0 MYA). The phylogenetic tree was graphically represented using MCMCTree v4.9f (http://abacus.gene.ucl.ac.uk/software/paml.html) (accessed on 1 September 2024) [45].

### 2.7. Analyses of Gene-Family Expansion and Contraction

We leveraged CAFE 5.0 [47] to investigate gene-family expansions and contractions, integrating data from identified gene families, a constructed phylogenetic tree, and estimated divergence times. Our approach utilized a stochastic birth and death model within the CAFE framework to trace the evolution of gene gains and losses across the phylogenetic tree. For each gene family, we computed conditional *p*-values, interpreting those below 0.05 as indicative of significant shifts in gene gain or loss rates. The analysis outcomes were graphically represented using iTOL (https://itol.embl.de/ (accession on 15 February 2025)) [48], offering a visual overview of the evolutionary dynamics of gene families across the taxa examined.

### 2.8. CAZyme and P450 Analysis

To annotate and classify the genes encoding CAZymes from the genomes of two *Auricularia* species and 26 other macrofungi, the CAZy database (http://bcb.unl.edu/dbCAN2/, accessed on 8 March 2023) was used with the HMMER 3.2.1 tool (filter parameter E-value < 10^−5^; coverage > 0.35). Bubble plots of CAZymes analyses were formed for these fungi using the complex heat map package in Rstudio v4.20.

The P450s and target protein sequences were predicted using Diamond 2.1.8 (E-value < 10^−5^) and Hmmer v3.3.2 (filter parameter: E-value < 10^−5^; coverage > 0.35). The P450s of all species were analyzed according to this standard.

### 2.9. Identification of Repetitive Elements and LTR Analysis

The identification of repetitive sequences in the genome was achieved through the utilization of RepeatMasker v4.0.5 [49] in conjunction with the Repbase TE library and the TE protein database. Subsequently, RepeatModeler [50] and LTR_FINDER v1.07 were utilized to predict the boundaries and family relationships of these sequences. Finally, Tandem Repeats Finder v4.0924 [51] was used to detect tandem repeat sequences. The results from these various tools were integrated, and redundant entries were removed to generate the final repeat annotation.

A de novo genome repeat library was constructed using RepeatModeler 2 v2.0.1 [50], which combines RECON v1.0.88 [52] and Repeat Scout v1.0.6 [51] for repeat discovery. The Repeat Classifier was then employed to classify the prediction results by searching against the Dfam v3.5 database [53]. LTR_FINDER v1.07. LTR_retriever v2.9.0 [54] was then used to create a high-quality, non-redundant library of complete LTRs. The sequences that flank the LTRs were extracted and aligned using MAFFT v7.205 [55], and genetic distances were calculated using the Kimura model in EMBOSS v6.6.0 [56].

The insertion time of the LTRs was determined using the formula T = K/(2r), where K represents the number of nucleotide substitutions per site between each pair of LTRs, and r is the nucleotide substitution rate, defined as the difference between LTRs divided by the substitution rate per year. All LTR fragments from full-length LTR-RTs were utilized as queries in a BLAST search (https://blast.ncbi.nlm.nih.gov/Blast.cgi (accessed on 1 September 2024)) against the genome sequence (E-value threshold: 1 × 10^−5^) to identify homologous fragments.

### 2.10. BGC Analysis and Visualization

BGCs prediction was performed using antiSMASH 7.0 [57], phylogenetic tree-based clustering using IQtree 2.2.3 [44], and correlation using specific parameters ‘-m MFP -bb 1000 -alrt 1000 -abayes -nt AUTO’ analysis. To further investigate the multi-domain enzyme NRPS, NRPS-like, RiPP, and Synthaser 1.1.22 [58] were used to further resolve their domains. These include domains like adenylation (A), thiolation (T), thioesterase (TE), condensation (C), and thioester reductase (TR).

### 2.11. Data Availability

The ITS sequence of *Auricularia* sp. Qinling M12 was registered in the NCBI GenBank under accession number OR702994.1, and the final genome assembly results and associated data have been submitted to NCBI under BioProject PRJNA1026359, BioSample SAMN37738836, and GenBank GCA_037042885.1, respectively. The ITS sequence of *Auricularia* sp. Qinling M13 was registered in the NCBI GenBank under accession number OR702993.1, and the final genome assembly results and associated data of the strain M13 have been submitted to NCBI under BioProject PRJNA1026369, BioSample SAMN37739492, and GenBank GCA_037042915.1, respectively.

## 3. Results

### 3.1. Resource Collection, Domestication, and Cultivation of Wild Auricularia Mushrooms in the Qinling Mountains

The Zhashui Auricularia industry is a prime example of the development of China’s Auricularia sector. In recent years, Zhashui Auricularia mushrooms have achieved remarkable progress [59]. The Qinling Mountains, with Zhashui County at their core, are notable for their abundant wild edible and medicinal mushroom resources, of which Auricularia mushrooms are a particularly salient example. During our team’s extensive exploration and utilization of wild macrofungi resources in the Qinling Mountains, we have amassed a substantial collection of freshly harvested and living *Auricularia* specimens. Through strain isolation and domestication cultivation of these samples, two strains, designated M12 and M13, have been identified as being significantly distinct from commercially available varieties in the market due to their distinct basidiocarps. A comprehensive evaluation has demonstrated that these two strains possess excellent and stable horticultural traits, including basidiocarp characteristics, yield, and growth cycle. They have been formally designated as *Auricularia* sp. qinling M12 (Figure 1A) and *Auricularia* sp. qinling M13 (Figure 1B). The strain M12 was collected from Jinmi Village in Zhashui County, while M13 was obtained from the Niubeiling National Forest Park, also located in Zhashui County. Given the status of the Qinling Mountains as the core production area for Zhashui *Auricularia* mushrooms and the pressing need for innovation in core germplasm resources, it is essential to conduct a genomic investigation of these two high-quality wild *Auricularia* strains originating from the Qinling Mountains.

### 3.2. Genome Sequencing, De Novo Assembly, and Annotation

K-mer analysis estimated *Auricularia* sp. qinling M12 is a dikaryon with 2.58% heterozygosity and about 55.20 Mbp genome size (Appendix A and Appendix A). The genome of the strain M12 was assembled using a multi-platform sequencing strategy. Totals of 34.70 Gbp of PacBio Sequel II HiFi reads (~619.18 × coverage), as well as 9.25 Gbp (~165.06 × coverage) of Hi-C data and 16.92 Gbp (~302.60 × coverage) of Illumina NovaSeq clean reads (Appendix A) were used for assembly, resulting in a 56.04 Mb genome (Figure 1C). After Hi-C-assisted assembly by ALLHIC, 56.04 Mb of genome sequence was mapped to 12 chromosomes (Figure 1D, Appendix A) and 13 unmapped contigs, with an N50 of 3,257,518 bp and a GC content of 57.01% (Appendix A). These chromosomes ranged in length from 2,451,749 bp to 9,151,808 bp (Figure 1C, Appendix A), totaling a 97.72% Hi-C anchoring rate. Among the 12 chromosoms, chromosome 12 (chr12) displays extensive homologous regions with chr1, alongside detectable partial homology with chr2, chr4, chr7, and chr8 (Figure 1C). A 254.88× average-depth, 99.91% coverage, and 93.1% of the BUSCOs (including 81.8% of the single-copy BUSCOs) indicated that the genome has good assembly completeness (Appendix A). K-mer analysis revealed a heterozygosity rate of 2.34% for strain M13, with a preliminary genome size estimation of 49.62 Mbp (Appendix A and Appendix A). Utilizing 5.39 Gbp of PacBio Sequel II HiFi pass reads combined with 5.82 Gbp of Illumina NovaSeq clean data, the final assembly yielded a refined genome size of 52.10 Mbp. This assembly comprises 14 contigs, achieving an N50 of 3,421,898 bp and exhibiting a GC content of 57.09%. Contig 1 represents the longest sequence, measuring 9,420,553 bp. In contrast, Contigs 13 (220,981 bp) and 14 (86,453 bp) are significantly shorter than the first 12 contigs, both falling below 0.3 Mb. The map-rate of 90.46%, average depth of 99.34×, coverage of 99.91%, and 93.8% BUSCO (including 89.4% single-copy BUSCO) indicated that the genome of M13 also had good assembly completeness (Appendix A).

The genome assembly of strain M12 contained 17,043 predicted protein-coding genes harboring 86,698 introns. The average transcript length was 1704.25 bp, with coding sequences (CDS) averaging 1272.25 bp. BUSCO assessment against the fungi_odb10 database revealed a complete score of 91.7% (78.6% single-copy, 12.9% duplicated), alongside 2.2% fragmented and 6.2% missing BUSCOs, indicating robust completeness and accuracy in gene prediction. Functional annotation was performed using nine databases: Nr, Pfam, eggNOG, UniProt, KEGG, GO, Pathway, RefSeq, and InterProScan. Of these, 15,425 genes (90.51%) received functional assignments, with the highest annotation rate in the NR database (15,235 genes; 89.39%) and the lowest in COG (1478 genes; 8.67%). Notably, 85.84% of NR-annotated genes showed homology to *A. subglabra* TFB-10046 S55. For strain M13, 16,484 protein-coding genes were predicted, containing 92,203 introns. BUSCO analysis demonstrated comparable prediction quality, with a complete score of 91.7% (85.1% single-copy, 6.6% duplicated). Employing the same annotation pipeline as M12, 14,597 genes (88.55%) were functionally annotated, again with NR providing the highest coverage (14,394 genes; 87.32%).

### 3.3. Comparative Genome Analysis Within Auricularia Species

Synteny analysis revealed extensive collinear relationships between the 12 chromosomes of strain M12 and the first 12 contigs of strain M13. Chromosome 8 of the strain M12 exhibited pronounced homology with Contig 1 of the strain M13. Large-scale syntenic blocks were identified between the following pairs: Chr2 and ctg2; Chr3 and ctg10; Chr4 and ctg7; Chr5 and ctg3; Chr6 and ctg4; Chr7 and ctg5; Chr9 and ctg8; Chr10 and ctg6; and Chr11 and ctg9 (Figure 2A). Genome-wide polymorphism profiling proves indispensable for functional gene mapping and genetic diversity studies. Through systematic analysis of Illumina NovaSeq sequencing data, 662,431 high-confidence SNPs were identified in strain M12, predominantly localized across its first 11 chromosomes (Figure 2B, Appendix A). Similarly, genomic investigation of strain M13 revealed 590,038 rigorously filtered SNPs (Figure 2C). Comparative genomic analysis was subsequently performed using *A. subglabra* TFB-10046 S55, the only publicly available annotated genome within the *Auricularia* genus, alongside the genomes of strains M12 and M13. Orthology assessment revealed 9028 conserved gene clusters shared among the three strains. Notably, *A. subglabra* TFB-10046 S55 possessed significantly more unique orthologous groups (1381) compared to both M12 (255) and M13 (278). Interspecific comparisons indicated markedly greater shared homologous genes between M12 and M13 (11,346) than between either strain and *A. subglabra* TFB-10046 S55 (M12: 9563; M13: 9552; Figure 2D,E).

In order to gain a deeper insight into the genomic characteristics of *Auricularia* strains M12 and M13 from Qinling Mountains, a comparative genomic analysis was conducted with five other *Auricularia* genomes. In terms of genome size, M12 and M13 (56.04 Mbp and 52.10 Mbp, respectively) were smaller than *A. subglabra* TFB-10046 SS-5 (74.92 Mbp), *A. cornea* ACW001-33 (78.72 Mbp), and CCTCCM 20221287 (73.17 Mbp), yet larger than *A. heimuer* Dai 13782 (49.76 Mbp) and A14-8 (43.57 Mbp). With respect to protein-coding gene content, M12 and M13 exhibited comparable gene counts to *A. heimuer* Dai 13782, slightly exceeded those of A14-8, and demonstrated significantly fewer genes compared to *A. subglabra* TFB-10046 SS-5 and *A. cornea* CCTCCM 20221287. Collectively, these metrics underscore the robustness of the sequencing and assembly quality for the M12 and M13 genomes (Table 1).

### 3.4. Phylogenetic and Gene Family Variation Analysis

To elucidate the phylogenetic placement and divergence chronology of the two *Auricularia* strains, M12 and M13, we reconstructed a phylogeny incorporating representative parasitic and saprotrophic basidiomycetes, with *Aspergillus oryzae* (Ascomycota) serving as the outgroup. The analysis employed 27,600 conserved single-copy orthologous proteins, revealing the following key estimates of divergence times between lineages based on molecular clock calibration. The crown age of the order Auriculariales was calculated at 114.665 MYAs with 95% highest posterior density (HPD) from 90.527 to 142.901 MYAs. Within the genus *Auricularia*, the strains M12 and M13 exhibited the closest evolutionary similarity, diverging 4.575 MYAs with 95% HPD from 3.253 to 6.178 MYAs. The divergence time between *A. subglabra* and the M12-M13 clade was estimated at 33.537 Mya (95% HPD: 24.611–44.261 Mya). Furthermore, the split between the *Auricularia* lineage and the clade comprising taxa Elmerina caryae and Exidia glandulosa occurred approximately 79.677 MYAs (95% HPD: 61.162–101.634 Mya) (Figure 3A). Further investigation using the reconstructed evolutionary trees revealed intricate patterns of gene contraction and expansion across 66,178 gene families in the genomes of these 29 species. Within the genome of *Auricularia* sp. Qinling M12, 608 out of 209 gene families were observed to have undergone expansion or contraction, and the strain M13, 330 out of 279 gene families were observed to have undergone expansion or contraction. Among the three species of the genus *Auricularia*, the most pronounced gene contraction and expansion was observed in *Auricularia* sp. Qinling M12 (Figure 3B). This suggests that *Auricularia* sp. Qinling M12 has experienced a significant amplification in gene families throughout its evolutionary history.

To further reveal the secondary metabolite synthesis capacity of the two *Auricularia* strains, M12 and M13, we correlated their CYP450 superfamily (Figure 3D,E). The results showed that the percentage of CYPs of fungi in different ecological niches had a large gap, and there was also a certain gap among fungi in the same ecological niche (Figure 3C, Appendix A). Among them, the strains M12 and M13 contained 150 and 153 CYPs, respectively, which were significantly lower than another species in the genus *A subglabra* TFB-10046 SS5. The results suggest that some of the CYPs genes in strain M12 and M13 may have underwent horizontal gene transfer (HGT) during the evolutionary process, which allowed the CYPs genes to acquire new functions to adapt to new environments, and at the same time reduced the diversity of secondary metabolites in the species to a certain extent.

### 3.5. CAZyme Analysis and Developing SSR Markers

Auricularia spp. are typical wood-rotting fungi that usually parasitize plant rhizomes and obtain nutrients for growth by breaking down the cellulose in plants. A comprehensive analysis of the genomes of *Auricularia* sp. Qinling M12 and *Auricularia* sp. Qinling M13 revealed that a total of 355 genes were used to encode 375 CAZymes in M12, while 344 genes encoded 371 CAZymes in M13. The CAZymes domain in strain M12 includes 216 glycoside hydrolases (GHs), two glycosyltransferases (GTs), 80 auxiliary activities (AAs), 41 carbohydrate esterases (CEs), 22 polysaccharide lyases (PLs), and 14 carbohydrate-binding modules (CBMs). While the azymes domain in strain M13 included 212 glycoside hydrolases (GHs), a glycosyltransferase (GT), 83 auxiliary activities (AAs), 35 carbohydrate esterases (CEs), 24 polysaccharide lyases (PLs) and 16 carbohydrate-binding modules (CBMs) (Figure 4A, Appendix A and Appendix A). The correlation results showed that two strains of the genus *Auricularia*, M12 and M13, had a high percentage of AAs to GHs in their CAZymes, which may be closely related to their living conditions. Notably, 17 CAZyme-encoding genes and 21 CAZyme-encoding genes with multiple functional structural domains were found in strain M12 and strain M13, respectively. The three genes g13884.t1, g384.t1, and g16089.t1 in strain M12 contained three structural domains, of which the two genes g13884.t1 and g384.t1 both had one GHs and two CBMs structural domains, whereas g16089.t1 contained by three GHs. In contrast, the six genes containing g12979.t1, g11069.t1, and g11263.t1 in strain M13 all contain three functional structural domains, with g11069.t1 consisting of three CEs structural domains. The diversity of functional structural domain compositions of CAZymes highlights to some extent the complexity and adaptability of these fungi in their ecological niche. Cluster analysis showed that strains M12 and M13 formed a branch with some members of Agaricales fungi. In terms of CAZyme characteristics, strains M12 and M13 were closest to *Oudemansiella raphanipes* CGG-A-s2 (Figure 4A).

Among them, SSRs are widely distributed in coding and non-coding regions of genes and play a key role in gene regulation and species evolution. Because of their significant polymorphism and wide distribution in the genome, they are often considered to be highly desirable molecular markers. By searching the M12 and M13 genomic sequences from scratch for SSRs with repeat units ranging from 1 to 6 nucleotides in length, a total of 4220 SSRs were identified in the 56.04 Mbp M12 genome, with a relative frequency of about 77 SSRs per Mb. In the 52.10 Mbp M13 genome, a total of 3755 SSRs were identified with a relative frequency of 75 per Mb, a slightly lower number than that of the M12 strain. However, it is interesting to note that the SSR composition of M12 and M13 strains had the highest percentage of trimer, and both of them accounted for 45.09% of the total number of SSRs. Comparative SSR analysis of seven fungi of the genus *Auricularia*, including strains M12 and M13, revealed that trimers were an important component of SSRs for most fungi of the genus, with a percentage generally greater than 40%. However, both strains *A. polytricha* MG66 and *A. subglabra* TFB-10046 had less than 40% trimers, with *A. subglabra* TFB-10046 having a relatively balanced proportion of monomers, dimers and trimers in its SSRs, whereas MG66 did not, with the largest proportion of monomers at 48.25% (Figure 4B, Appendix A).

### 3.6. TE Analysis and Genome Duplication

Repeat sequences are an essential component of the genome, accounting for a significant proportion of the genome. Molecular tools based on these repetitive sequences play an important role in genetic breeding, varietal identification, and genetic mapping markers. Repetitive sequences, also known as transposable elements (TEs), are mainly of four types: short interspersed nuclear elements (SINE), long interspersed nuclear elements (LINE), long terminal repeats (LTR), and DNA transposable elements (DNA-TE). In addition, in the sequencing annotation of this experiment, LTRs include Gypsy, Copia, and unclassified LTRs. *Auricularia* sp. Qinling M12 was annotated by genome sequencing to obtain a total of 21,842 repetitive sequence units with a total length of 77,158,804 bp, accounting for 14.13% of the genome. Among them, the highest number of LTRs accounted for 5.19% of the total genome. In addition, there were 8563 unknown types of repetitive sequences, accounting for 8.95% of the total genome. There were 20,276 repetitive sequence units in strain M13, with a total length of 7,237,846 bp, accounting for 13.89% of the total genome. Similar to strain M12, LTRs were also the most dominant component, accounting for 7.03% of the total genome, and their positional type of repetitive sequences accounted for a lower percentage than that of M12, accounting for only 7.92% of the total genome (Figure 5A, Appendix A). The insertion time of LTR was analyzed for strains M12 and M13, and a continuous insertion of LTR was observed for both strains since nearly 2 MYA. M12 is in the peak region, and for M13, although the peak LTR insertion density can be observed, it is still near the peak, and neither of them has completed the full insertion of LTR (Figure 5B).

Furthermore, detailed TE analysis for six fungi, including M12 and M13, showed that the latest insertion time of intact TEs into the genomes of these species was almost 0 MYA (Figure 5C). At the subtype level, the Gypsy and Copia-LTR elements in M12 and M13 are right around the peak of the LTR amplification peak. Given the prevalence of Gypsy and Copia-LTR in TEs, it is hypothesized that the M12 and M13 genomes are recently undergoing a massive TE amplification in the genome (Figure 5B).

To gain insight into the impact of repetitive sequences on the genome, whole-genome duplication (WGD) analyses were conducted. The non-synonymous substitution rates (Ka), synonymous substitution rates (Ks), and their ratios (Ka/Ks) were calculated for homozygous gene pairs. The main peak positions of the ratios of Ka and Ks of the three Auricularia, including *Auricularia* sp. Qinling M12 and M13, were less than 1, and the peak shapes were narrower, indicating that the gene functions of these three strains were highly conserved and the selective pressures were more homogeneous (Figure 5D, Appendix A). Compared with the two strains M12 and M13, the peaks of *A. subglabra* TFB-10046 SS5 were lower and broader, indicating that the purifying pressure it experienced was more decentralized, which might be related to the fact that M12 and M13 experienced the continuous insertion of LTR.

### 3.7. Search and Analysis of Secondary Metabolite-Related Genes

Considering the high nutritional and medicinal value of *Auricularia* mushrooms, a genome search and analysis of genes (clusters) related to the biosynthesis of its secondary metabolites was performed. The genomes of two strains, M12 and M13, were predicted using AntiSMASH. The prediction results showed that the M12 genome contained a total of 21 gene clusters, encoding 30 core genes, including six terpene synthesis-related enzymes, 11 NRPS-like enzymes, ten ribosomally synthesized and post-translationally modified peptide (RiPP) enzymes, and three Indole enzymes (Table 2). These 21 BGCs were distributed on ten chromosomes, with the highest number of core genes on Chr9 with nine genes and the lowest number of core genes on Chr1, Chr6, Chr10, and Chr11 with only one core gene (Figure 6A, Table 2). In contrast, the M13 genome contains 30 gene clusters that encode 38 core genes, including nine terpene synthesis-related enzymes, ten NRPS-like enzymes, 14 RiPP enzymes, three Indole enzymes, and two isocyanide (Table 2). These 30 BGCs were distributed on ten chromosomes, with the highest number of core genes on ctg8 with 12 genes and the lowest number of core genes on ctg5, ctg9, ctg10, and ctg11 only one core gene (Figure 6B, Table 2).

These genes were further analyzed because the core genes play a crucial role in the synthesis of secondary metabolites. Considering that sesquiterpenes are the major active components of macrofungi, we further analyzed the terpene synthases annotated in strains M12 and M13, and screened a total of 13 sesquiterpene synthases out of the 15 terpene synthases predicted to be obtained by Antismash (Appendix A). To further explore the pattern of pyrophosphate farnesyl cyclization catalyzed by these sesquiterpene synthases, the researchers performed an evolutionary tree-based cluster analysis using these 13 STSs with a resulting collection of 429 STSs of fungal origin (Appendix A). Categorizing the 13 STSs from M12 and M13 sources with three different branches, the clade of sesquiterpene synthases characterized by 1,6-cyclization of (3R/S)-NPP cyclization had up to eight enzymes and the enzymes in M12 and M13 usually appeared in pairs, suggesting to some extent that terpene production was not affected by the divergence of the two strains (Figure 6C). Antismash prediction revealed that there were no PKS-encoding genes in strains M12 and M13, and a total of 21 NRPS, NRPS-like encoding genes were collected from the genomes of these two fungi. Multi-structural domain analysis of these 21 genes revealed that M13_g12874.t1, M13_g9211.t1, and M12_g9872.t1 had multiple C domains, A domains, and T domains, and the structural domains showed a high degree of consistency. This suggests that these genes may be used to encode the formation of cyclized multimers. Furthermore, the M12 and M13 strains contain a variety of NRPS, NRPS-like coding genes, which, to some extent, determine the diversity of nonribosomal peptide compounds of this class of strains (Figure 6D).

## 4. Discussion

The Qinling Mountains, which form an important ecological barrier in western China, are characterized by high forest cover and diverse vegetation, making them a natural habitat for fungi, especially macrofungi. These conditions provide unique advantages for the growth of macrofungi, resulting in a rich diversity of species. In recent years, several new species of macrofungi have been discovered in the Qinling region [62,63]. Our research team has focused on exploring macrofungal germplasm resources in the Qinling Mountains and has successfully isolated strains such as *Laetiporus sulphureus* [64], *Cryptoporus qinlingensis* [65], *Cyathus olla* [66], *Hericium rajendrae* [67], and *Trichaptum biforme* [68]. Among these, valuable edible and medicinal fungi such as *C. qinlingensis* [65] and *H. rajendrae* [67] have been identified. The Qinling Mountains span the province of Shaanxi and serve as the boundary between central and southern Shaanxi. The southern mountainous region of Shaanxi, located in the heart of the Qinling Mountains, is a key area for the edible mushroom industry in Shaanxi Province. In 2023, Shaanxi Province produced 128.83 tons of edible mushrooms, a year-on-year growth of 0.93%, with black mushrooms accounting for 15.20% of the total production, making it one of the main pillars of the province’s edible mushroom industry [24]. Zhashui County, located in the core area of the Qinling Mountains in southern Shaanxi, has established wood ear mushrooms as a cornerstone of its local economy, earning it the nickname “Golden wood ear” among local residents [25]. The two strains studied here, M12 and M13, were isolated from the core region of Qinling, which is closely associated with Zhashui County. The results of this study are expected to provide important resources for the further development and utilization of wood ear mushroom germplasm in this region.

Although the genomes of several species in the genus *Auricularia* have been reported previously [22,60,69], the overall quality of these genome assemblies has been suboptimal. In this study, we used HI-C technology for the first time to achieve a high-precision genome assembly for strain M12, successfully anchoring the genome to 12 chromosomes. This resulted in a final genome size of 56.04 Mb, with an N50 length of 3.26 Mb and a BUSCO completeness of 93.1%. Although the assembly quality of another strain, M13, did not reach the chromosomal level, it was ultimately assembled into 14 scaffolds, resulting in a high-quality genome with a size of 52.10 Mb, an N50 length of 4.34 Mb, and a BUSCO completeness of 93.8%. These results have gone some way to addressing the lack of high-quality genomes for species within the genus *Auricularia*, and have provided a solid foundation for future research in this genus. Subsequent comparative genomic analyses between the two *Auricularia* strains from the Qinling Mountains and other basidiomycetes revealed that M12 and M13 are most closely related. In particular, their CYP450 content is significantly lower than that of the congeneric strain TFB-10046 SS5, which, to some extent, confirms the similarity of their habitats. The closest relatives outside the genus *Auricularia* are the species *E. caryae* and *E. glandulosa*, which diverged from the *Auricularia* species approximately 79.677 MYAs. Based on the results of LTR insertion time analysis, it is hypothesized that these two lineages have undergone large-scale TE expansion during their evolutionary history, which may be associated with their adaptation to changing environmental conditions. Comparative analyses of CAZymes in 24 basidiomycetes, including strains M12 and M13, revealed that *Auricularia* species exhibit high levels of AA and GH domains. This is likely linked to their ecological niche as wood-decay fungi, as higher AA and GH functional domains enable stronger lignin and cellulose degradation capabilities to meet their nutritional requirements. Additionally, this study provides the first detailed investigation of SSRs among *Auricularia* species. The identification of diverse SSRs across different strains offers an informative foundation for the development of genetic molecular markers.

Macrofungi often hold significant culinary and/or medicinal value, as exemplified by the highly revered Reishi mushroom (Ganoderma lucidum) [70,71]. These fungi, which serve both culinary and medicinal purposes, embody the concept of food and medicine being interconnected [72,73]. The genus *Auricularia* is a valuable medicine–food homology fungus in China, known for its rich nutritional and pharmacological properties. Studies have demonstrated that polysaccharides from *Auricularia auricula-judae* can improve glucose metabolism in diabetic mice by regulating the AKT/AMPK signaling pathways and modulating gut microbiota composition [17]. Additionally, these polysaccharides have been shown to enhance energy metabolism and metabolic health in obese mice [18]. These nutritional and biological activities are closely associated with secondary metabolic pathways in *Auricularia* species. For instance, using antiSMASH for the analysis of secondary metabolite biosynthesis gene clusters in M12 and M13, we identified 21 and 30 BGCs (biosynthetic gene clusters), respectively. The core gene types were predominantly RIPPs, indicating the potential of these strains to synthesize diverse bioactive peptides. This finding provides theoretical guidance for the development of bioactive compounds from *Auricularia* species.

## 5. Conclusions

This study constitutes the inaugural genomic sequencing and high-quality assembly of two strains of *Auricularia* species from the Qinling region, which exhibit desirable horticultural traits. The genome of strain M12, sized at 56.04 Mbp, was anchored to 12 chromosomes using HI-C technology, while strain M13, with a genome size of 52.10 Mbp, was assembled into 12 scaffolds. Comparative analyses, incorporating collinearity assessment, CYP450 quantification, CAZymes classification and quantification, and gene family expansion/contraction studies, revealed a close genetic relationship between the two strains, with a divergence time of approximately 4.575 MYAs. In contrast, their divergence from *A. subglabra* TFB-10046 SS5 was estimated to be much earlier, at 33.537 MYAs. LTR insertion time analysis indicated that these strains are currently undergoing large-scale TE expansion. Analysis of secondary metabolite biosynthesis revealed that the genomes of M12 and M13 encode 21 BGCs, including core genes for terpenoids and NRPS, and 30 BGCs with 38 core genes, respectively. Overall, this study provides two high-quality genomes of *Auricularia* species, establishing a robust foundation for understanding their genetic evolution and facilitating biotechnological applications such as molecular breeding.

## Figures and Tables

**Figure 1 jof-11-00395-f001:**
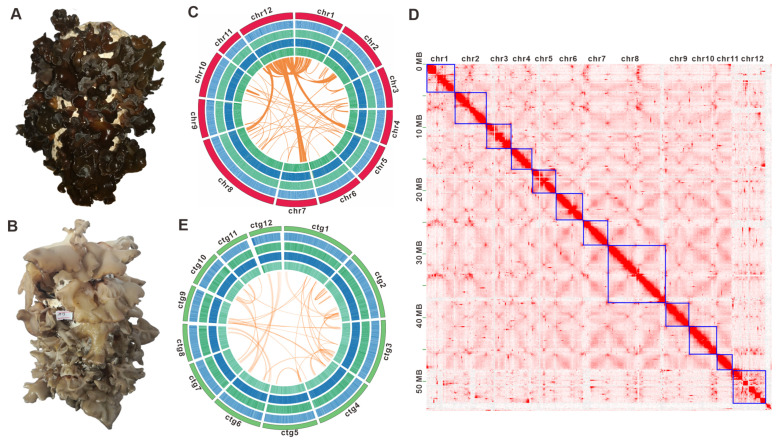
Morphological and genomic features of *Auricularia* sp. Qinling M12 and *Auricularia* sp. Qinling M13. (**A**) The fruiting body of *Auricularia* sp. Qinling M12. (**B**) The fruiting body of *Auricularia* sp. Qinling M13. (**C**,**E**) Genomic characterization of *Auricularia* sp. Qinling M12 and *Auricularia* sp. Qinling M13. From the outside to the inside are I. Chromosome and contigs; II–IV. GC-density, GC-skew, AT-skew (window size 1 kb); V. Gene-density (window size 1kb), collinear regions between and within chromosomes (contigs). Chr, chromosomes. (**D**) The Hi-C interaction map of *Auricularia* sp. Qinling M12. The numbered chromosomes serve as coordinates, with the color of each dot indicating the log value of interaction intensity between corresponding bin pairs of the genome. The interaction intensity increases from white (low) to red (high). Chr, chromosomes.

**Figure 2 jof-11-00395-f002:**
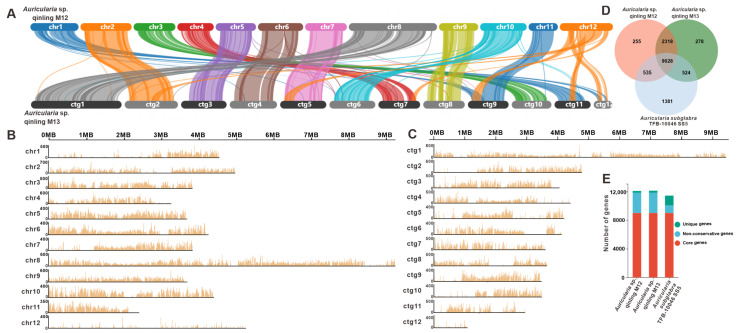
Comparative genomic analysis of *Auricularia* sp. Qinling M12 and M13. (**A**) Collinearity analysis of *Auricularia* sp. Qinling M12 and M13. (**B**,**C**) SNP analysis of *Auricularia* sp. Qinling M12 and M13, respectively. (**D**) Venn schematic of comparative genomes within the genus *Auricularia*. (**E**) Schematic histogram of comparative genomes within the genus *Auricularia*.

**Figure 3 jof-11-00395-f003:**
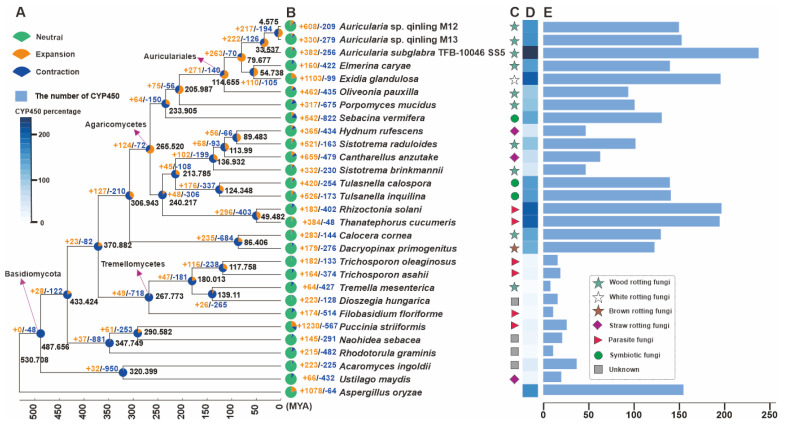
Phylogeny and gene family variation. (**A**) The evolutionary relationship and expanded and contracted gene families among *Auricularia* species and 26 representative Basidiomycetes. The maximum likelihood method credibility tree was inferred from 203 single-copy orthologous genes. All nodes received full bootstrap support. The divergence time is labeled as the mean crown age for each node, and the black numbers at the branches indicate the corresponding divergence times in MYA. (**B**) The proportion of expansion and contraction in the genomes. (**C**) Ecological niche of each species. (**D**) Heatmap of percentages of CYP450 members. (**E**) The gene numbers of CYP450 members in genomes.

**Figure 4 jof-11-00395-f004:**
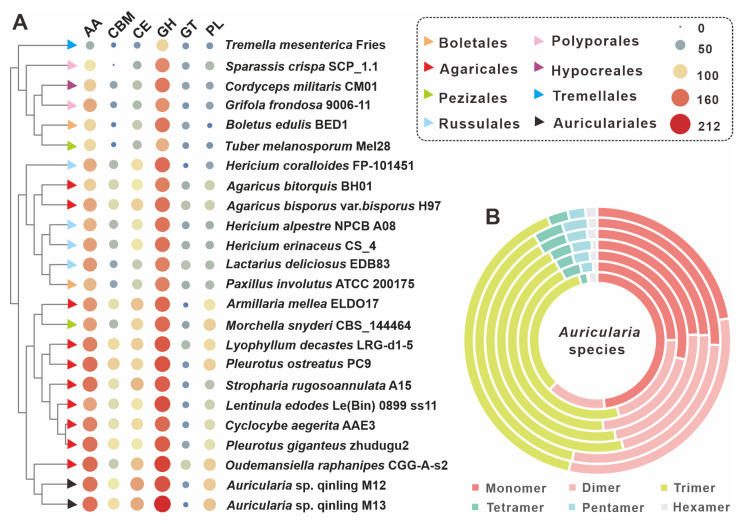
CAZymes analysis of *Auricularia* and related edible mushrooms. (**A**) Composition comparison of CAZymes among 24 edible fungi, including *Auricularia* sp. Qinling M12 and *Auricularia* sp. Qinling M13. (**B**) Relative abundance of seven types of SSRs in the genus *Auricularia*. From the outside to the inside are *A. cornea* CCMJ2827, *A. subglabra* TFB-10046, *Auricularia* sp. Qinling M12, *Auricularia* sp. Qinling M13, *A. heimuer* Dai 13782, *A. auricula-judae* B14-8, and *A. polytricha* MG66.

**Figure 5 jof-11-00395-f005:**
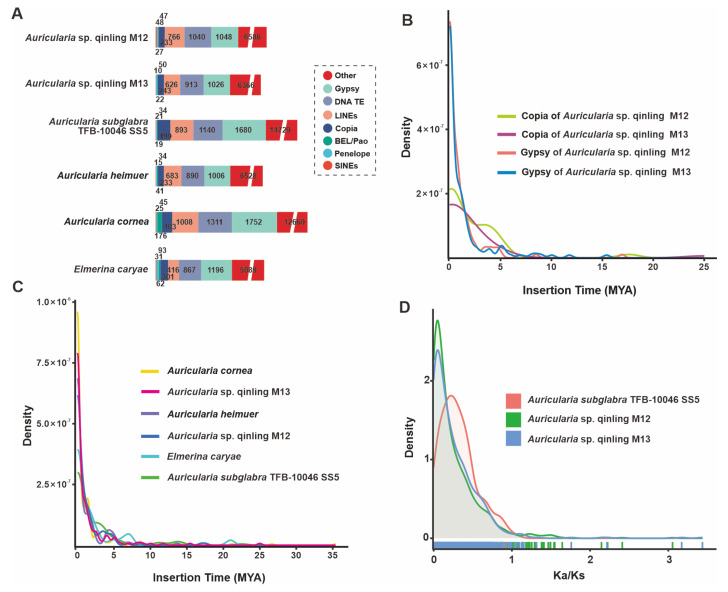
Analysis of TEs and positively selected genes in the *Auricularia* sp. Qinling M12 and M13 genomes and four closely related taxa. (**A**) Comparison of TE families in the six taxa. (**B**) Insertion bursts of Gypsy and Copia elements in *Auricularia* sp. Qinling M12 and M13. (**C**) Comparison of temporal patterns of intact LTR-RT insertion bursts in the six taxa. (**D**) Frequency distributions of Ka/Ks between homologous gene pairs of the three taxa.

**Figure 6 jof-11-00395-f006:**
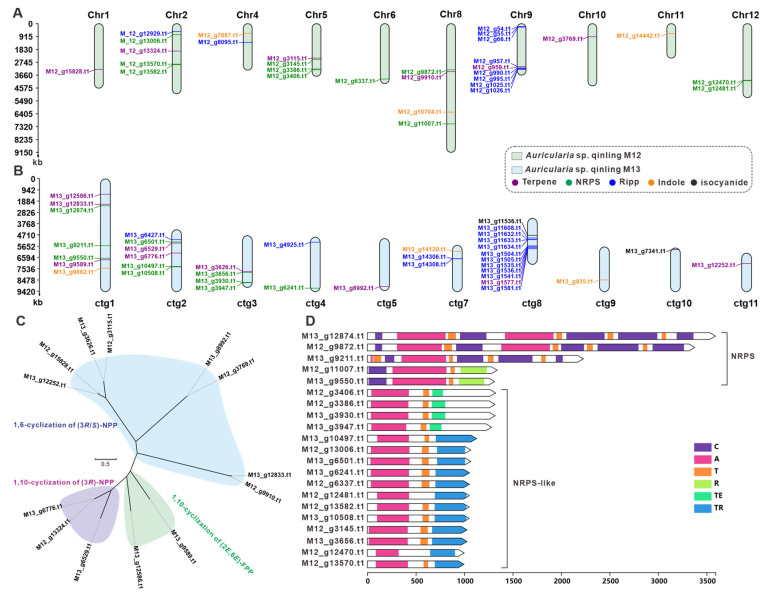
The core genes involved in secondary-metabolite biosynthesis from *Auricularia* sp. Qinling M12 and M13. (**A**,**B**) Distribution of biosynthetic core genes for natural products on the chromosomes and contig, (**C**) phylogenetic tree analysis for STSs, (**D**) domain characterization of the core enzymes containing multiple domains.

**Table 1 jof-11-00395-t001:** Genomic comparison within *Auricularia* species.

Entry	*Auricularia subglabra* TFB-10046 SS-5	*Auricularia heimuer* Dai 13782	*Auricularia cornea* ACW001-33	*Auricularia heimuer* A14-8	*Auricularia cornea* CCTCCM 20221287	*Auricularia* sp. Qinling M12	*Auricularia* sp. Qinling M13
Sequencing technology	Sanger, Roche, Illumina	Illumina, PacBio	Illumina, PacBio	Illumina	Illumina, PacBio	Illumina, Nanopore	Illumina, Nanopore
Sequencing depth	46.6×	56.0×	150.0×	101×	101.0×	254.88×	99.34×
Genome size (bp)	74,920,202	49,761,846	78,720,327	43,569,032	73,170,506	56,039,961	52,096,326
No. of contig	4688	114	23	NA	NA	13	14
No. of scaffold	1531	103	NA	535	86	NA	NA
No. of chromosome	NA	NA	13	NA	NA	12	NA
Largest length (bp)	2,214,130	2,880,034	8,224,108	964,551	10,960,066	7,410,467	9,420,553
Contig N50 (bp)	36 kb	1,350,668	5,662,720	269,985	5,486,495	3,257,518	4,342,189
BUSCO completeness (%)	98.89	92.40	98.53	99.08	92.6	93.1	93.8
GC content (%)	58.5	56.98	59.5	57.09	59.09	57.01	57.09
No. of protein-coding genes	23,555	16,244	18,574	14,094	19,120	17,043	16,484
GenBank accession No.	GCA_000265015.1	GCA_002287115.1	GCA_030578095.1	GCA_002092955.1	GCA_041684135.1	GCA_037042885.1	GCA_037042915.1
Reference	Dimitrios Floudas, et al. [20]	Yuan Yuan, et al. [21]	Xiaoxu Ma, et al. [60]	Ming Fang, et al. [61]	Lei Ye, et al. [22]	This study	This study

NA indicates not available.

**Table 2 jof-11-00395-t002:** Putative biosynthetic gene clusters responsible for secondary metabolites in the genome of *Auricularia* sp. Qinling M12 and M13.

Species	Cluster No.	Location	Start (bp)	End (bp)	Core Gene ID	Core Gene Type
*Auricularia* sp. Qinling M12	1	Chr1	3,253,766	3,285,838	M12_g15828	Terpene
	2	Chr2	518,487	600,102	M12_g12929	RIPP
	3	Chr2	745,074	805,878	M12_g13006	NRPS-like
	4	Chr2	1,929,993	1,965,389	M12_g13324	Terpene
	5	Chr2	2,855,306	2,948,006	M12_g13570 M12_g13582	NRPS-like
	6	Chr4	658,839	687,964	M12_g7887	Indole
	7	Chr4	1,291,307	1,377,608	M12_g8093	RIPP
	8	Chr5	2,436,420	2,458,591	M12_g3115	Terpene
	9	Chr5	2,505,971	2,576,983	M12_g3145	NRPS-like
	10	Chr5	3,201,511	3,311,118	M12_g3386 M12_g3406	NRPS-like
	11	Chr6	3,908,077	3,970,050	M12_g6337	NRPS-like
	12	Chr8	3,248,971	3,320,951	M12_g9872	NRPS
	13	Chr8	3,396,683	3,431,424	M12_g9910	Terpene
	14	Chr8	6,266,874	6,298,324	M12_g10704	Indole
	15	Chr8	7,112,735	7,174,514	M12_g11007	NRPS-like
	16	Chr9	174,483	296,954	M12_g54 M12_g55 M12_g66	RIPP
	17	Chr9	3,029,649	3,118,695	M12_g957 M12_g959	RIPP Terpene
	18	Chr9	3,131,265	3,311,684	M12_g990 M12_g995 M12_g1025 M12_g1026	RIPP
	19	Chr10	900,829	930,335	M12_g3769	Terpene
	20	Chr11	713,973	738,997	M12_g14442	Indole
	21	Chr12	3,999,942	4,092,373	M12_g12470 M12_g12481	NRPS-like
*Auricularia* sp. Qinling M13	1	Ctg1	1,263,880	1,283,376	M13_g12586.t1	Terpene
	2	Ctg1	2,089,154	2,104,999	M13_g12833.t1	Terpene
	3	Ctg1	2,228,179	2,274,164	M13_g12874.t1	NRPS
	4	Ctg1	5,551,760	5,590,378	M13_g9211.t1	NRPS
	5	Ctg1	6,615,406	6,656,749	M13_g9550.t1	NRPS
	6	Ctg1	6,742,657	6,761,802	M13_g9589.t1	Terpene
	7	Ctg1	7,464,731	7,478,093	M13_g9862.t1	Indole
	8	Ctg2	785,232	846,699	M13_g6427.t1	RIPP
	9	Ctg2	1,002,177	1,045,104	M13_g6501.t1	NRPS-like
	10	Ctg2	1,107,535	1,117,659	M13_g6529.t1	Terpene
	11	Ctg2	1,938,121	1,956,815	M13_g6776.t1	Terpene
	12	Ctg2	3,064,296	3,134,023	M13_g10497.t1 M13_g10508.t1	NRPS-like
	13	Ctg3	2,969,089	2,983,152	M13_g3626.t1	Terpene
	14	Ctg3	3,043,959	3,086,421	M13_g3656.t1	NRPS-like
	15	Ctg3	3,883,715	3,909,922	M13_g3930.t1	NRPS-like
	16	Ctg3	3,923,290	3,962,473	M13_g3947.t1	NRPS-like
	17	Ctg4	390,206	441,980	M13_g4925.t1	RIPP
	18	Ctg4	4,226,330	4,264,556	M13_g6241.t1	NRPS-like
	19	Ctg6	3,984,555	4,004,024	M13_g8992.t1	Terpene
	20	Ctg7	501,819	507,204	M13_g14120.t1	Indole
	21	Ctg7	1,105,012	1,162,775	M13_g14306.t1 M13_g14308.t1	RIPP
	22	Ctg8	1,397,656	1,436,913	M13_g11536.t1	isocyanide
	23	Ctg8	1,609,754	1,660,555	M13_g11608.t1	RIPP
	24	Ctg8	1,738,874	1,786,304	M13_g11632.t1 M13_g11633.t1 M13_g11634.t1	RIPP
	25	Ctg8	2,287,310	2,344,999	M13_g1504.t1 M13_g1505.t1	RIPP
	26	Ctg8	2,374,982	2,441,370	M13_g1535.t1 M13_g1536.t1 M13_g1541.t1	RIPP
	27	Ctg8	2,503,216	2,557,232	M13_g1577.t1 M13_g1581.t1	Terpene RIPP
	28	Ctg9	2,737,821	2,759,584	M13_g935.t1	Indole
	29	Ctg10	123,365	153,057	M13_g7341.t1	isocyanide
	30	Ctg11	839,405	858,175	M13_g12252.t1	Terpene

## Data Availability

The original contributions presented in this study are included in the article/Appendix A. Further inquiries can be directed to the corresponding authors.

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
