# Peer review of "Genomic Sequencing and Characterization of Two Auricularia Species from the Qinling Region: Insights into Evolutionary Dynamics and Secondary Metabolite Potential"

_jof, 2025, doi:10.3390/jof11050395_

Round 1
Reviewer 1 Report
Please see the attachment.
Please see the attachment.

Author Response
The reviewed manuscript is related to specific aspects of the general problem of more efficient commercial production of edible and medicinal mushrooms. To develop the medicinal components, the theoretical basis for the subsequent genetic breeding, as well as functional gene mining, provide an important genomic resource. The edible and medicinal Auricularia mushroom in China is mainly produced in the Qinling region. The authors study two wild Auricularia strains selected from the Qinling region, present sequencing assembly results, and analyze the secondary metabolite-associated biosynthetic gene clusters.
Q1. The structure of Introduction section should be modified. Within Introduction, several items have to be highlighted related to (i) the main question addressed by the research; (ii) a specific gap in the field addressed by the work; (iii) achievements of the previous published material in the given subject area, (IV) the manuscript goal(s) in brief conceptual form at the end of Introduction, as usual.
A1: We appreciate your insightful and constructive comments. We are too inclined to agree with your comments. As you can see now, we have added a paragraph at the end of the introduction section. This text has perfectly addressed your concerns. In fact, we did have some content during the manuscript preparation process, but it was lost when we transferred the text to the JOF template.
Q2. Materials and Methods, Lines 244-248: there are two repeated fragments, both related to the strain M12, however included different digits for BioProject, BioSample and GenBank. One of two fragments might come to the strain M13, please check.
A2. We sincerely appreciate the reviewer's careful reading and valuable comments. After re-reading the manuscript, we confirm that there is indeed an omission of repeated information in lines 244-248. The second fragment (BioProject: PRJNA1026369, BioSample: SAMN37739492, and GenBank: GCA_037042915.1) should correctly correspond to strain M13 instead of M12, and this error has been corrected. We have checked the BioProject, BioSample and GenBank numbers of both strains against the NCBI database several times to ensure accuracy. Thank you again for bringing this important issue to our attention.
Q3. The abbreviation SNP should be spelled out the first time it is used in the text (Line 170) in the form "Single-Nucleotide Polymorphism (SNP).
A3. Thank you for your careful review, and we have revised line 170 to explicitly define the term "single nucleotide polymorphism (SNP)" in the first mention of it.
Q4. The authors wrote that the comparative genomic analysis was performed using A. subglabra TFB-10046 S55, which is "the only publicly available annotated genome within the Auricularia genus (Line 327). However, "a comparative genomic analysis was conducted with five other Auricularia genomes that were available" (Line 337). Furthermore, the information on A. subglabra TFB-10046 SS5 was properly included in the section 2.4 "Comparative Genomic Analysis", but not on the other 4 strains of the fungus, A. cornea (ACW001-33 and CCTCCM 20221287), and A. heimuer (Dai 13782 and A14-8). Please explain.
A4:Thank you for your fair assessment. These descriptions can indeed be quite perplexing. Here’s the actual situation: there are five published and accessible genomes of the genus Auricularia, including A. subglabra TFB-10046 S55. However, of these five, only A. subglabra TFB-10046 S55 comes with annotation files (protein sequences, coding region sequences, and gff3 files). It’s only the genome files with protein sequences that can be used for gene family analysis and evolutionary position analysis based on orthologous single-copy genes. The other four genomes have only genome sequence files, which can only be used for analyses related to genome assembly quality. We’ve made some appropriate revisions to the text to ensure it’s informative, scientific, and clear.
Q5. Section 3.3: "M12 and M13 (XX Mbp and XX Mbp, respectively)" (Line 339). Does it mean that it is an erroneously made record to be changed into "M12 and M13 (56.04 Mbp and 52.10 Mbp, respectively)"? Please check.
A5. We sincerely thank the reviewers for their meticulous attention to detail in identifying this potential error. The values mentioned in line 339 'M12 and M13 (XX Mbp and XX Mbp, respectively)' are actually placeholder text that was inadvertently left in the manuscript during the final editing stage. As correctly pointed out by the reviewers, these values should be replaced with the exact genome sizes of 56.04 Mbp (M12) and 52.10 Mbp (M13). We have carefully checked the entire manuscript to ensure that all values and placeholders have been removed or replaced with validated data.
Q6. "analysis employed XXX conserved single-copy orthologous proteins (Line 358): please check.
A6: Revised and thank you.
Q7. The section 3.6 begins with "Considering the high nutritional and medicinal value of Mucuna pruriens, a genome search and analysis …". That seems to be error again, please check.
A7: We sincerely appreciate the reviewer’s keen observation and critical feedback. The statement in section 3.6 ('Considering the high nutritional and medicinal value of Mucuna pruriens ......') was indeed an unintentional error introduced during the writing of the manuscript and should have read Auricularia mushrooms. We have corrected this error and have reviewed the entire manuscript to correct this type of error.
Q8. "were collected from the genomes of these two bacteria" (Line 531). What bacteria are told? Please explain.
A8: Thank you for your insightful feedback and for pointing out this error in the text. After reviewing the manuscript, 'these two bacteria' in line 531 is a typographical error and actually refers to strains M12 and M13, the focus of our research in this paper, and should be described as these two fungi rather than two bacteria.
Q9. There are no titles, any notes or explanations within the Supplementary Material files of the type "Dataset S1-3.xlsx".
A9: Added and thank you.
Q10. Please check Supplementary Material, Dataset S1, section "M13", and change language into English.
A10: We thank the reviewer for bringing this issue to our attention. Upon review of the Supplementary Material (Dataset S1, section "M13"), we confirm that a portion of the content was inadvertently written in Chinese due to an oversight in document formatting. We have now fully translated this section into English and rigorously proofread it to ensure linguistic accuracy and consistency with the rest of the manuscript.
Q11. Please check Supplementary Material, Dataset S3, and insert names of the columns, which are totally absent in this file.
A11. We are truly grateful to the reviewers for their meticulous attention and apologies for the oversight in the original submission. Upon re-examining Supplementary Material Dataset S3, we have confirmed that the column headings were inadvertently omitted during the final file conversion. This error has now been rectified to ensure the data’s complete transparency and reproducibility.
Reviewer 2 Report
The work is showing an impresive volume of information, however the structuration of the content should be slightly modified since parts missing in Introduction and MM section are within Results and discussion section. SOme sections should increase the detail (specially MM) and the clarity should be improved.
The word is lacking of an hypothesis or objectives, the strains should be identified at species level at least a putative species or species group should be identified. Having the genomes sequenced you can use ITS, 16S or 23S to blast and have an aproximated idea. And also you should submit this sequence to genbank related to your strains to make it clearer and traceable.
L 50-55 The redaction is confussed. Explain better and straightforward that Auricularia heimuer is the correct name of the species used in China for long time. Avoiding to consider the Auricularia auricula-judae the consumed in China if it was not in fat the species used in China. Is the Heimuer one species or is a group of species, it should be clear.
L 57”… after the Chinese phonetic alphabets of Heimuer” it is not clear, did you mean after transcribing phonetically the Chinese word 黑默尔.
L 59 Which one is the accepted species A. cornea or A. nigricans? I suggest to just use the accepted one and avoid to refer to the synomim.
L 68 The Introduction is quite short, and more detail should be convenient, At the end of the Introduction an extra paragraph must be added to point the hypothesis or the objective of the work.
Materials and Methods
L 87 Which strain, of which species and obtained from where? You should clearly identify the strain including if was bought or isolated and the identity species, strain code and if available ITS sequence.
L 92 “Genome Size and Complexity Investigation:” Delete this words which are not related with the paragraph and is not in the same format of tittles and subtitles.
L 99 and 105 If you want to add subsections add in the format 2.2.2 and 2.2.3
The entire subsection 2.1 is lacking any reference and can be introduced from the strain used (if previously used in other work) to the library construction methods.
L 124 Add more details to “ …using CEGMA 2.5[23] and BUSCO 4.0[24].”
L137-138 Add reference for reference genomes: “Auricularia subglabra TFB-10046 SS5 and Auricularia heimuer” The article were they were used or the accession of the database.
L 163 It is the first reference of the strains M12 and M13. Are your own strains? Are the same species or different? Identify previously the strains in section 2.1 and include it in the abstract.
L 169 Which species?
192 “2.6”
L 252-258 It should be part of the Introduction adding the corresponding reference.
259-264 It looks like Material and Methods. I suggest to move there, if not you should add background of how the isolation was done. Precise collection place, how you maintained the stains in culture, details of the comprehensive evaluation. Which tests you did to ensure the horticultural traits of these Auricularia sp. Even more, if the basidiocarps are different, how you know that these strains are Auricularia? You should analyze at least the ITS or 16S sequences.
L 264-268 Reformulate this sentence to be the objective of your work, and place it at the end of Material and Methods section
L 272 Only genome of M12 was sequenced using multi-platform sequencing strategy? What about M13? Did you also sequenced M13 genome, right? And how?
L 297 The figure 1 footnote is not explaining properly the panels C,D and E, which are also small in the figure and hard to interpret. Split the figure in different figures, separating the morphological images A and B fro the genomic figures. Add more details in the footnote and enlarge the figures.
L 312-313 Don´t refer here to the pipeline, explain clearly the pipeline in the material and methods section and identify which samples were analyzed with which pipeline.
L 320 The Chr and ctg are not properly explained in the manuscript, and should be declared in Introduction and/or Material and Methods section. Why the M12 you analyzed chromosomes and in M13 analyzed contigs? You should compare equivalents contigs with contigs or chromosomes with chromosomes.
L 326 Which genomic investigation? You declared several analysis in MM identify which analysis reveal the SNPs. And even more how you can ensure the SNPs if you are not comparing with other individuals of the same species?
L 330 why you declare that you have 3 species if you don´t know the identity of M12 and M13? You don´t know at all how many species have or you didn´t declared it alongside the manuscript.
L 337 “species”
L349 Figure 2 is too small is hard to see each subpanel and extract information from the draws in it.
L 358 What is “… XXX conserved…” you can use either phylogenetic oy phylogenomic approach to build a tree, but you should identify the genes or regions used for the reconstruction.
L 359 “key divergence estimates”? What do you mean? Do you refer to stimates of divergence times between lineages?
L 359 What do you mean by “Crown”? you refer to diversification or divergence time?
L 362 Evolution is not expressed as affinity, change it by similarity.
L 363 It is not the mean divergence, but it is the divergence between the A. subglabra with the clade M12-M13.
L 377 The figure is small and should be enlarged. The B subpanel is extremely hard to see. The C subpanel is informative and relevant to your work? And D and E subpanels are based on CYP450 members which is named here for the first time in the manuscript, and should be clearly explaind in Material and Methods section to help the reader to understand the work.
L 380 Show the bootstrap values and declare the support threshold (>70).
L 390 the strain M13 is here referred as Qinling M13 while in the rest of the manuscript is just referred as M13. Be consistent.
L 393 Did you performed any analysis of HGT?
L494 Mucuma pruriens??? Unitl this line your article was related to Auricularia species, why you are referring here to Mucuna pruriens?
L 544 Should be interesting refer in the Introduction and or MM section to the germplasm. Referring to previous works in Introduction or if you did in this work detailing it in MM.
L 545 This should be explained in MM section not here.
L 567-568 How it was? You analyzed both strains in the same way? Have you an explanation? Whe I was reading MM and Results section it was not clear to me.
And why you consider that 12 segments of 56 Mb with a completedness of 93.1% are chromosomes while 14 segments of 52 Mb with a completedness of 93.8% are contigs. To me looks pretty similar in number, lenghth and completedness to be considered both as the same either contigs or chromosomes.
Author Response
The work is showing an impresive volume of information, however the structuration of the content should be slightly modified since parts missing in Introduction and MM section are within Results and discussion section. Some sections should increase the detail (specially MM) and the clarity should be improved.
Reply: We are immensely grateful for the time and effort you have devoted to reviewing our manuscript and for your perceptive comments. Your recognition of the significance of our work is much appreciated, and we are also thankful for highlighting areas where our manuscript falls short. Your feedback will undoubtedly enhance the quality of our manuscript. Specific revisions are detailed in the subsequent comments.
The word is lacking of an hypothesis or objectives, the strains should be identified at species level at least a putative species or species group should be identified. Having the genomes sequenced you can use ITS, 16S or 23S to blast and have an aproximated idea. And also you should submit this sequence to Genbank related to your strains to make it clearer and traceable.
Reply: We are extremely grateful for your constructive feedback and wholeheartedly agree with your perspective. Reviewer 1’s comments are in line with your observations. The omission occurred when we transferred the original manuscript to the JOF template, inadvertently leaving out a crucial paragraph. This has now been added to the final paragraph of the Introduction. Thank you once again for your considered insights.
The ITS sequences of the two strains have also been submitted to GenBank, and the corresponding accession numbers have now been added to the manuscript.
Q1. The tittle doesn´t refer to which species or how many were analyzed, neither which are the insights from the genomic comparisson.
Reply: The species in question are from the genus Auricularia, which is clearly indicated in the title. The number of species has now been added. The genomic comparison is a subsequent aspect of the genomic analysis, and its insights are reflected in the dynamics mentioned in the title. Thank you for your comment and alternative perspective.
Q2. The Introduction is quite short, and more detail should be convenient, At the end of the Introduction an extra paragraph must be added to point the hypothesis or the objective of the work. There are parts of the manuscript which are in the Results and discussion section which can be included within introduction.
Reply: I couldn't agree more with your comments.The missing descriptions have been added as requested.
- In line 50-55 The redaction is confussed. Explain better and straightforward that Auricularia heimuer is the correct name of the species used in China for long time. Avoiding to consider the Auricularia auricula-judae the consumed in China if it was not in fat the species used in China. Is the Heimuer one species or is a group of species, it should be clear.
Reply: Modified and revised.
- In line ”… after the Chinese phonetic alphabets of Heimuer” it is not clear, did you mean after transcribing phonetically the Chinese word “黑木耳”.
Reply: Done.
- In line 59 Which one is the accepted species A. cornea or A. nigricans? I suggest to just use the accepted one and avoid to refer to the synomim.
Reply: Among the wild and cultivated species of Auricularia in China, Maomuer (毛木耳), is primarily composed of two species within the genus Auricularia: A. cornea and A. polytricha, as shown by molecular identification. A. cornea and A. polytricha are distinct species within the genus Auricularia. This is clearly stated in the revised manuscript. Thank you for your careful consideration."
- In line 68 The Introduction is quite short, and more detail should be convenient, At the end of the Introduction an extra paragraph must be added to point the hypothesis or the objective of the work.
Reply: added and thank you.
Q3. The methods are not adequately described, the authors hidde the inforation about how the strains were obtained and preserved, it makes untraceable the work. The authors didn`t declare the name of the strains in subsection 2.1 making hard to follow the work. It is not clear at all how M12 and M13 strains were analyzed, how were treated in the same way (or not), and why with similar results were interpret as different (chromosomes of M12 vs contigs of M13). More detail is needed in some parts of the manuscript and better correspondence between declared within MM and what is presented in Results is needed.
Reply: Thank you for your detailed review, the relevant details have now been added to Section 2.1. In fact, these details were touched upon in Section 3.1, as the reviewer pointed out. Of course, the revised manuscript has now been enhanced to include this information more fully, making it more objective and scientifically robust.
Q4. The redaction of the results should be improved to be considered as clear, some details are missing, the correspondence with MM sections should be clarified.
Reply: Thank you for your detailed review.
- In line 87 Which strain, of which species and obtained from where? You should clearly identify the strain including if was bought or isolated and the identity species, strain code and if available ITS sequence.
Reply: Added.
- In line 92 “Genome Size and Complexity Investigation:” Delete this words which are not related with the paragraph and is not in the same format of tittles and subtitles.
Reply: Done as requested.
- In line 99 and 105 If you want to add subsections add in the format 2.2.2 and 2.2.3
Reply: Thank you for your suggestion.
- The entire subsection 2.1 is lacking any reference and can be introduced from the strain used (if previously used in other work) to the library construction methods.
Reply: Thank you for your suggestion. This is the first report on the study of these two strains, and there are no directly relevant references available. The necessary descriptions for library construction have been added.
- In line 124 Add more details to “ …using CEGMA 2.5[23] and BUSCO 4.0[24].”
Reply: Done.
- In line 137-138 Add reference for reference genomes: “Auricularia subglabra TFB-10046 SS5 and Auricularia heimuer” The article were they were used or the accession of the database.
Reply: Added.
- In line 163 It is the first reference of the strains M12 and M13. Are your own strains? Are the same species or different? Identify previously the strains in section 2.1 and include it in the abstract.
Reply: The strains M12 and M13 are described in the abstract, Section 2.1,and Section 3.1.M12 and M13 are distinct species, as clearly outlined in Section 3.1.
- In line 169 Which species?
Reply: Revised and thank you.
- In line 252-258 It should be part of the Introduction adding the corresponding reference.
Reply: Thank you for your suggestion. The relevant content has now been incorporated into the concluding paragraph of the Introduction. Maintaining these descriptions as they are doesn’t feel redundant or jarring.
- Line 259-264 It looks like Material and Methods. I suggest to move there, if not you should add background of how the isolation was done. Precise collection place, how you maintained the stains in culture, details of the comprehensive evaluation. Which tests you did to ensure the horticultural traits of these Auricularia Even more, if the basidiocarps are different, how you know that these strains are Auricularia? You should analyze at least the ITS or 16S sequences.
Reply: Thank you for your detailed analysis. The relevant details have been added. The basidiocarps of M12 and M13 are indeed distinct, as shown in Figures 1A and 1B. The ITS sequences confirm their identification as Auricularia species, and the corresponding accession numbers have been added to the manuscript.
- L 264-268 Reformulate this sentence to be the objective of your work, and place it at the end of Material and Methods section.
Reply: Thank you for your valuable suggestions. We have thoroughly revised the manuscript in accordance with your comments.
- L 272 Only genome of M12 was sequenced using multi-platform sequencing strategy? What about M13? Did you also sequenced M13 genome, right? And how?
Reply: Both M12 and M13 were sequenced using PacBio Sequel II HiFi sequencing combined with Illumina NovaSeq sequencing.M12 was further scaffolded to the chromosome level with Hi-C data. In contrast, M13 was not subjected to Hi-C sequencing and thus was assembled only at the contig level. This is clearly described in Section 3.2.
Q5. The figures should be enlarge and the footnotes should be improved to be really informative.
Reply: modified and revised.
- line 297 The figure 1 footnote is not explaining properly the panels C, D and E, which are also small in the figure and hard to interpret. Split the figure in different figures, separating the morphological images A and B from the genomic figures. Add more details in the footnote and enlarge the figures.
Reply: The figure legends have been thoroughly refined and expanded.The enlarged Figure 1 now clearly displays the details of panels A and B.Given the correlation between panels 1A and 1C(and 1B and 1D),presenting these results together in a single figure is more persuasive.Thank you for your suggestion.
- L349 Figure 2 is too small is hard to see each subpanel and extract information from the draws in it.
Reply: Revised.
- L 377 The figure is small and should be enlarged. The B subpanel is extremely hard to see. The C subpanel is informative and relevant to your work? And D and E subpanels are based on CYP450 members which is named here for the first time in the manuscript, and should be clearly explaind in Material and Methods section to help the reader to understand the work.
Reply: P450s are important functional genes in macrofungal genomes and are crucial for understanding their genomic architecture. The methods for their analysis have been added to Section 2.8. The C subpanel, which illustrates the ecological niches of common macrofungi including Auricularia mushrooms, helps to elucidate the similarities and differences between these fungi and other macrofungi. This is also one of the common presentations in fungal genomic reports. In the revised version, high-quality vector graphics have been provided, which clearly display detailed information.
Q6. L 312-313 Don´t refer here to the pipeline, explain clearly the pipeline in the material and methods section and identify which samples were analyzed with which pipeline.
Reply: The descriptions present the results of the BSCUO analysis and the comparative outcomes, rather than the analytical process itself. The corresponding analytical process is detailed in Section 2.2. Thank you for your alternative interpretation.
Q7. L 320 The Chr and ctg are not properly explained in the manuscript, and should be declared in Introduction and/or Material and Methods section. Why the M12 you analyzed chromosomes and in M13 analyzed contigs? You should compare equivalents contigs with contigs or chromosomes with chromosomes.
Reply: The assembly results of fungi can be scaffolded to the chromosome level based on Hi-C data, which is a well-established and widely accepted analytical method in the field. In contrast, the genome of M13 was assembled solely using PacBio Sequel II HiFi sequencing combined with Illumina NovaSeq sequencing, which typically results in contig-level assemblies. Please be aware of this distinction. Thank you.
Q8. L 326 Which genomic investigation? You declared several analysis in MM identify which analysis reveal the SNPs. And even more how you can ensure the SNPs if you are not comparing with other individuals of the same species?
Reply: We sincerely appreciate the reviewer’s insightful comments. The genomic survey method used in this study aligns with the aforementioned approach, specifically employing SNP prediction, with the detailed workflow comprehensively documented in Section 2.5 of the Materials and Methods.
Q9. L 330 why you declare that you have 3 species if you don´t know the identity of M12 and M13? You don´t know at all how many species have or you didn´t declared it alongside the manuscript.
Reply: We sincerely thank the reviewer for raising this critical point. The taxonomic status of strains M12 and M13 was determined through ITS sequence analysis, confirming their classification within the genus Auricularia. Complete experimental protocols for ITS amplification and bioinformatic alignment parameters are provided in Section 2.11 of the Materials and Methods. We emphasize that the current identification resolves taxonomic placement at the genus level but does not satisfy the criteria for novel species designation, necessitating revision of the original "three species" terminology to "three strains" for precision (Line 393). This terminology specifically encompasses the investigated strains (M12 and M13) along with the publicly annotated A. subglabra TFB-10046 S55 (NCBI Genome Assembly ID: GCA_000265015.1), the latter serving as the sole reference taxon with validated genomic metadata in this genus.
Q10. L 337 “species”
Reply: We have revised the designation from "species" to "strains".
Q11. L 358 What is “… XXX conserved…” you can use either phylogenetic oy phylogenomic approach to build a tree, but you should identify the genes or regions used for the reconstruction.
Reply: We sincerely appreciate the reviewers' diligent examination of our manuscript. The placeholder identifiers (e.g., "XXX") in question were inadvertently retained during manuscript preparation. These have now been rectified with validated numerical values through comprehensive verification. A full audit of the document has been completed, ensuring all provisional markers have been replaced with experimentally derived data.
Q12. L 359 “key divergence estimates”? What do you mean? Do you refer to stimates of divergence times between lineages?
Reply: The term “divergence estimates” in the original text specifically refers to ​​divergence time estimates between lineages​​, calculated using molecular clock analysis. We have made changes here in the original manuscript. (Line 421).
Q13. L 359 What do you mean by “Crown”? you refer to diversification or divergence time?
Reply: The term 'crown age' is a technical term referring to the time when the most recent common ancestor of all extant species within a particular taxonomic group appeared. In this context, the crown age of the order Auriculariales refers to the time when the most recent common ancestor of all extant species within the order Auriculariales appeared.
Q14. L 362 Evolution is not expressed as affinity, change it by similarity.
Reply: Done.
Q15. L 363 It is not the mean divergence, but it is the divergence between the A. subglabra with the clade M12-M13.
Reply: Corrected.
Q16. L 380 Show the bootstrap values and declare the support threshold (>70).
Reply: A16.We reconstructed the phylogenetic tree using the best-fit model identified by IQ-TREE, and the bootstrap values are contingent upon the model employed. However, regardless of the model, the optimal model will yield the highest bootstrap support. Thus, it is typically described as “full.”
Q17. L 390 the strain M13 is here referred as Qinling M13 while in the rest of the manuscript is just referred as M13. Be consistent.
Reply: Done.
Q18. L 393 Did you performed any analysis of HGT?
Reply: HGT is a speculative description here. We have made the revision.
Q19. L494 Mucuma pruriens??? Unitl this line your article was related to Auricularia species, why you are referring here to Mucuna pruriens?
Reply: We express profound gratitude for the reviewers' insightful observations and constructive critique. We acknowledge an inadvertent taxonomic misassignment in the original manuscript where the specification should correctly read Auricularia mushrooms. And We have made corrections to the original text.
Q20. L 544 Should be interesting refer in the Introduction and or MM section to the germplasm. Referring to previous works in Introduction or if you did in this work detailing it in MM.
Reply: The relevant information has been added at the end of the Introduction section.
Q21. L 545 This should be explained in MM section not here.
Reply: Revised.
Q22. L 567-568 How it was? You analyzed both strains in the same way? Have you an explanation? Whe I was reading MM and Results section it was not clear to me.
Reply: The analysis methods for both strains were identical. In a genomic report paper that involves comparative content, such an approach is mandatory.
Q23. And why you consider that 12 segments of 56 Mb with a completedness of 93.1% are chromosomes while 14 segments of 52 Mb with a completedness of 93.8% are contigs. To me looks pretty similar in number, lenghth and completedness to be considered both as the same either contigs or chromosomes.
Reply: Similar issues have arisen on numerous occasions. Firstly, the chromosomes of M12 were obtained through Hi-C data analysis, whereas M13 did not undergo Hi-C sequencing. The level of assembly has no direct correlation with parameters such as BSCUO, and this should be noted.